# Sterba's Problem of Evil vs. Sterba's Problem of Specificity: Which Is the Real Problem?

Michael S. Jones

College of Arts & Sciences, Liberty University, Lynchburg, VA 24515, USA; msjones2@liberty.edu

**Abstract:** In 2019 the noted ethicist and political philosopher James Sterba published a new deductive version of the argument from the problem of evil to the conclusion that an Anselmian God does not exist. In this article I will argue that Sterba's argument involves a problematic sorites-type paradox that, in order to be consistent, he should view as undermining his argument, since in his previous work on ethics he viewed this same sort of problem as counting as a significant objection to moral cultural relativism. I show how his arguments involve a sorites-like paradox, explain how this is damaging to the argument from evil, and conclude by offering suggestions for how Sterba might address this weakness.

**Keywords:** God; horrendous evil; James Sterba; Pauline Principle; problem of evil; problem of specificity; sorites paradox; theodicy

## 1. Introduction

One of the most persistent and most effective arguments against belief in God is the argument from the "problem of evil" (henceforth POE). From Epicurus' day to ours, atheists have used the existence of evil as empirical evidence that an omniscient, omnipotent, and omnibenevolent God does not exist. And for just as long, theists have been offering responses to these arguments.[1] In 2019 the noted ethicist and political philosopher James Sterba published *Is a Good God Logically Possible?*, adding his weight to the ranks of philosophers who argue that evil counts against the existence of God (Sterba 2019). In this book he develops a deductive version of the argument from evil that he believes conclusively demonstrates that an omniscient, omnipotent, and omnibenevolent God is not logically possible.[2]

In his work on ethics Sterba utilizes an argument against moral cultural relativism. This argument hinges on the existence of ambiguity in moral cultural relativism that he thinks renders it incoherent (Sterba 2013, pp. 23, 25). My argument, in short, is that this ambiguity results from an implicit sorites-type paradox inherent in moral cultural relativism and that the same sort of ambiguity exists in Sterba's argument from evil. Therefore, either Sterba must revise/reject his argument against moral cultural relativism, or he must revise/reject his argument from evil. Since his argument against moral cultural relativism seems strong to me, I believe he should do the latter.

## 2. Sterba's Argument

I will begin by explaining Sterba's argument from the POE. Sterba's main areas of work are moral and political philosophy.[3] His work in this area is well known and highly respected.[4] In recent years, he has also begun working on issues in philosophy of religion.[5] He believes that certain strategies that have proven effective in ethics and political philosophy can be usefully employed in philosophy of religion as well.[6] The particular issue in philosophy of religion that seems to have captured his attention is the POE. Over the last decade he has organized two conferences at the University of Notre Dame on the topic, published an anthology on the POE, published several articles on various aspects of

the POE, and as his own argument from evil to the nonexistence of God has developed, he has made presentations on it at many universities and conferences, including the American Philosophical Association and the Society for Philosophy of Religion. Both his book and its argument are the result of an intense period of research and productivity.[7]

Consistent with this strategy of applying lessons drawn from moral and political philosophy to issues in philosophy of religion, Sterba wants to apply to the POE an axiom that is sometimes called the "Pauline Principle." This is an ethical principle relating to the Principle of Double Effect (Sterba 2019, pp. 2ff and 49–66). The Pauline Principle, which is named after the Apostle Paul, states that it is not moral to perform an evil act for the purpose of bringing about good.[8] This principle forms a premise of one step in Sterba's argument. This step argues that it is not moral for God, if he exists, to perform an evil act for the purpose of bringing about good. For ease of reference, I will call this argument from the Pauline Principle the PPA (short for the Pauline Principle Argument).

Sterba acknowledges that there are exceptions to the Pauline Principle. He grants at least three: it is moral to perform an otherwise evil act (1) if the evil is trivial, (2) if it is easily reparable, or (3) if it is the only way to prevent a far greater evil (Sterba 2019, p. 3). Because of these exceptions, he focuses on the problem of the existence of "significant evils" that an omnipotent God could prevent but has not.[9] He focuses on significant evils precisely because he wants to use the Pauline Principle to show that if an Anselmian God (one who is omniscient, omnipotent, and omnibenevolent) were to exist, then some kinds of evil that exist would not exist, and he recognizes that lesser evils may not be sufficient for this task. This distinction between lesser evils and significant evils is important to my argument: he does not appear to take the existence of lesser evils as evidence for the nonexistence of God.[10] Only significant evils sufficiently ground the PPA.

After introducing the PPA, Sterba moves on to anticipate various possible theistic responses. In successive chapters he discusses and responds to the Free Will Defense (chp. 2), the Irenaean or "soul-making" theodicy (chp. 3), skeptical theism (chp. 5),[11] the argument that God is not a moral agent (chp. 6), the greater good theodicy (also chp. 6, beginning on 125), and the redemption history theodicy (chp. 7).[12] In his conclusion he briefly addresses attempts to respond to the POE by limiting God's power or holiness (Sterba 2019, p. 192). Since my objection to his argument is not dependent on the success of any of these attempts to respond to the POE, I will not take the time to discuss his attempts to repudiate them. Other philosophers have analyzed the strengths and possible weaknesses of Sterba's contributions to each of these and their analyses have been published in this and other periodicals.[13]

More salient to my planned objection are certain other nuances that Sterba introduces as he develops his argument. Perhaps the most significant of these is made during his discussion of the greater good theodicy in chapter six (specifically pp. 126–30). Here, he points out that there are exactly four types of goods that might possibly justify God in causing or permitting evil: first-order goods to which we are entitled, first-order goods to which we are not entitled, second-order goods to which we are entitled, and second-order goods to which we are not entitled.[14] He argues that these four categories exhaust all of the possible goods that could justify God in causing or permitting some specific evil. He believes that if the goods that fall into these categories are unable to justify God in causing or permitting evil, then the greater good type of theodicy fails, since there are no goods except those that fall into one of these four categories.

Here, I must explain these four categories of goods. Sterba does not clearly define them: he seems to believe that what he means will become apparent as he discusses them. The impression I get is that by "goods to which one is entitled" he means goods to which one has a right and that it would be immoral to take from someone. Examples of these, at least according to our founding fathers, might be life, liberty, and the pursuit of happiness. Of course, under certain circumstances it may be moral to take these from someone, so these examples are not completely uncontroversial, but in any case Sterba takes it that there are things to which people are entitled. These fall into two subcategories: first-order

goods and second-order goods. First-order goods to which one may be entitled are goods to which one has a right that do not depend on the existence of some serious wrongdoing, such as the right to freedom from brutal assault (Sterba's example). Second-order goods to which one may be entitled are goods to which one has a right that do depend on the existence of some serious wrongdoing, such as a victim's right to aid from brutal assault (again Sterba's example) (Sterba 2019, pp. 126–28).

By "goods to which one is not entitled," he seems to mean goods that one does not have a right to expect or demand and that others are not morally obligated to bestow. He does not provide examples of such goods in *Is a Good God Logically Possible?*, but perhaps examples would include all luxury items and any superabundance of items that in more modest amounts are necessities. These fall into the same two subcategories as the goods to which one is entitled: first-order and second-order. First-order goods to which one is not entitled are goods to which one does not have a right and that do not depend on the existence of some serious wrongdoing, such as a luxury car and expensive meals. Second-order goods to which one is not entitled are goods to which one does not have a right that do depend on the existence of some serious wrongdoing, such as stolen goods or life-saving organs bought on the black market (Sterba 2019, pp. 129–30).[15]

Returning to the argument: Sterba believes that if the goods that fall into these four categories are unable to justify God in causing or permitting evil, then the greater good type of theodicy fails. Furthermore, he argues that for an omnipotent being, permitting evil is morally equivalent to causing it. According to the PPA, it is not moral to perform an evil act in order to bring about good results (with the possible exceptions noted above). Thus for Sterba, the PPA also proscribes permitting preventable evil acts in order to bring about good results. Therefore, he concludes that it is not moral for God to cause or permit evil in order to bring about any of the goods described above except possibly within the limits ascribed to the PPA: when the evil is trivial, easily reparable, or the only way to prevent a far greater evil. Consequently we see once again that causing or permitting significant evils is very difficult to justify.

One notable aspect of Sterba's argument is that he has eschewed the trend toward inductive "evidential" forms of the POE in favor of a deductive argument. In fact, he seems to view himself as fixing the problems with the deductive version of the argument developed decades ago by the Australian philosopher J.L. Mackie (Sterba 2019, 25ff.).[16] So let me lay out Sterba's argument deductively. It appears to be a modus tollens:

- Major Premise: If there is an omnipotent, omnibenevolent God, then that God would neither cause nor permit significant evil to happen.
- Minor Premise: It's not the case that God neither causes nor permits significant evil to happen.
- Conclusion: Therefore it is not the case that there is an omnipotent, omnibenevolent God.[17]

This appears to be a deductively valid argument.[18]

A second notable aspect of Sterba's argument is its roots in moral and political theory. In the introduction to his book he makes much of the idea that there are resources in moral and political philosophy that could fruitfully be brought to bear on issues in philosophy of religion. Others have applied lessons from metaphysics and epistemology to issues in philosophy of religion with great success, so I am inclined to think that this idea is a good one. His attempt to do this using the Pauline Principle is a good illustration of the possibilities of this endeavor. I would like to make my own attempt at this, taking another lesson from ethics and applying it to the PPA.

## 3. My Argument

Sterba's book on the POE is not the only place where he makes use of the Pauline Principle. He also uses it in his ethical writings; for example, on page 57 of his book *Introducing Ethics for Here and Now* he explains the Pauline Principle and discusses its

possible use as an objection to Utilitarian ethics.[19] Earlier in the same book he introduces another principle, one that is often called "the problem of specificity"(Sterba 2013, p. 23). This "problem" forms the basis of one of Sterba's objections to moral cultural relativism.[20] It is this principle that I will attempt to apply to Sterba's PPA to show that it fails.

Sterba's objection to moral cultural relativism is that if morality is relative to cultural groups, then the term "cultural groups" must have a concrete referent; i.e., cultural groups must exist in order for morality to exist. However, it's not clear that the term has a concrete referent: is morality relative to nations—each nation creating and abiding by its own set of moral norms? However, within large nations like the US there are many subcultures that have mores that are distinct from the mores of other American subcultures. Should we say that morality is relative to such subcultures? Perhaps we should, but if we do, then we must reckon with the fact that many people belong to more than one subculture. And so the analysis goes, moving to ever smaller and smaller subsections of society until one arrives at individual moral subjectivism instead of cultural moral relativism. That is why the problem of specificity is a challenge to cultural relativism. As Sterba puts it: "Any act (e.g., contract killing) could be wrong from the point of view of some particular society (e.g., US citizens), right from the point of view of a subgroup of that society (e.g., the Mafia), and wrong again from the point of view of some particular member of that society or other subgroup (e.g., law enforcement officers). However, if this were the case, then obviously it would be extremely difficult for us to know what we should do, all things considered" (Sterba 2013, p. 23).

The problem of specificity is a sort of sorites paradox. This is a philosophical problem attributed to the ancient Greek philosopher Eubulides of Miletus (Oms and Zardini 2019, p. 3). "Sorites" is the Greek word for a pile or a heap, and the paradox can be illustrated by a heap of sand: 10,000 grains of sand sitting on the floor in front of us would clearly constitute a heap of sand. If we remove one grain, what is left would still be a heap. The same is true if we remove another grain. However, if we continue to remove grains of sand, one at a time, eventually the pile would become so small that we would no longer say that there is a heap of sand on the floor. The oddity is that it does not seem like removing a single grain of sand would render a heap of sand a non-heap, and we probably would not be able to identify any single grain the removal of which marked the point when the heap turned into a non-heap, but yet somewhere along the way the heap a non-heap does become.

This paradox applies to more than just physical collections of objects. Another common illustration of the paradox has to do with age: there is a very significant and very clear difference between person A at age 18 and person A at 80. When we see her now, after 80 years of life and labor, we recognize that she is old. If we saw her last week, she would still look old. If we saw her two weeks ago—or three weeks, or four weeks—she would look old to us. If we continue to subtract weeks, one at a time, we may never discover a particular week where she goes from appearing old to appearing not old. Yet if we subtract weeks, one at a time, long enough, we will arrive at a point where she is in fact very young—first 18, and then even younger, and this in spite of the fact that subtracting any specific week is an insufficient condition to make her young.

What is "old" and what is "young," and why is it so difficult to find a clear demarcation between the two? Hrafn Asgeirsson, in an article titled "The Sorites Paradox in Practical Philosophy," points out that part of the issue here is terminological vagueness (Asgeirsson 2019, pp. 229–45). If language were such that every word has a fixed and very precise meaning, then perhaps the problem would not exist. If "heap" was universally accepted as meaning "any collection of four or more objects stacked upon each other" (and if "old" for humans was universally understood to refer to people who are in the last third of the current human life expectancy) then perhaps there would be no sorites paradox. However, that is not how natural languages work. Nor is it how concepts work, and the real issue probably has more to do with our concepts than the labels that we give to them (see Asgeirsson 2019, pp. 321–22).

A sorites paradox occurs when there is no clear conceptual point at which we go from Σ to Σ′, no clear point where we go from not being a heap to being a heap. Such a situation could arise because no such point exists, or it could be that such a point exists but is not known. If the latter is the case, then Asgeirsson points out two further options: the transition point may be unknown but knowable or it may be unknown because it is unknowable (Asgeirsson 2019, pp. 231–33). Various scholars have opted for each of these possibilities, and as of yet there does not seem to be a consensus on which is the most accurate assessment of the problem.

Obviously, the reason that this is called a "paradox" is that while it is clear that there is a difference between a few grains of sand upon the floor and a heap of sand on the floor, we are unable to identify when the collection ceases to not be a heap and actually becomes one. I certainly do not feel old, but my granddaughter clearly views me as old. Am I old? The evidence supports both that I am and that I am not. Hence the paradox.

The "problem of specificity" objection to moral cultural relativism points out this vagueness in cultural relativism and uses it to show that moral cultural relativism is wrong. Sterba seems to endorse this argument. Hence he seems to accept that such vagueness is a significant problem for a philosophical position. I think that the same sort of vagueness appears in his version of the POE. Let me explain.

Sterba argues that significant evils are incompatible with the existence of an Anselmian God. However, he grants that the existence of evils that are trivial, easily reparable, or the only way to prevent a far greater evil may be compatible with such theism. This seems to imply that there is a distinguishable difference between theism-compatible evils and those evils that he labels "significant," the evils that are incompatible with theism.

How does he distinguish between these? He does not say. He does give us several examples of "significant evil," though. One of these is parents permitting their children to be brutally assaulted as a means of building character (Sterba 2019, p. 57). Perhaps an example of trivial evil would be allowing someone to steal candy from your child in order to teach him or her a lesson about sharing. Both incidents cause the child pain. In fact, a child who has his or her candy taken away can feel rather intense emotional pain. I do not know if it would rival the intensity of the pain of being brutally beaten, and I am sure it would not do the long-term damage that knowing that your parents allowed you to be brutally beaten would do. However, the comparison involves us in considering a spectrum of child-pain-inducing events and judging that some are trivial and others significant. How do we determine which is which? Can we objectively draw a line to the left of which everything is trivial and to the right of which everything is significant? It is not clear to me how we should set about doing so.

This is a problem, for some evils that person A might experience as trivial might seem very significant to person B. If person B has only experienced trivial evils, then medium evils might seem like horrendous evils to him. On the other hand, if person A has experienced tremendous evils, then medium evils might seem trivial to her.

Sterba's PPA entangles us in a sorites-type paradox. He claims that some evils are trivial and some are significant, that the latter are not compatible with the existence of an Anselmian God, that the latter exist, and that therefore an Anselmian God does not exist. However, he has not and perhaps cannot tell us what evils are significant enough to disprove God's existence. His argument seems to assume that we already know that significant evils exist, but perhaps the evils that we have experienced actually fall on the trivial side of the spectrum.

It seems at least possible that God is already preventing numerous evils that are even worse than the most significant evils that we now witness. What appear to us to be significant evils may be much less significant than the evils that God is preventing. Sterba could argue that God can and should prevent both those evils that are more significant than the ones that we experience and the ones that we actually do experience. However, if God were to do that, then the evils that are slightly less evil than the most significant evils that we're currently experiencing would seem to us to be the truly significant evils and

thus in need of God's intervention. This, I suspect, would be true every time God prevents a lesser set of evils, since what we take to be significant evil is probably relative to other things that we are experiencing.

Sterba grants that the existence of small evils is compatible with the existence of an Anselmian God. Because Sterba has embroiled us in a sorites-type paradox, there is no clear point at which an evil ceases to be insignificant. Therefore, on Sterba's argument, there is no clear point at which an evil ceases to be compatible with God's existence. Hence the conclusion that any particular evil to which Sterba can point is incompatible with God's existence seems unjustified. This calls into question the minor premise of the syllogism given above.

### 4. Conclusions

In this article I have attempted to show that because Sterba thinks that a sorites-type paradox fatally undermines moral cultural relativism, he should view the existence of a sorites-type paradox in his deductive version of the POE as undermining his argument from the POE. At this point, however, I believe it would be hasty and rash to simply conclude that I have defeated Sterba's argument. I know him to be a creative and resourceful thinker, and he may have a way around my objection. Furthermore, I myself can see several potential paths that he could try in response to my attempted critique.

One option for saving Sterba's argument would be for him to repudiate his "problem of specificity" argument against moral cultural relativism and argue that vagueness is a ubiquitous feature of human discourse that does not present a serious problem to philosophical arguments and theories. Other philosophers have already championed this position, and it would enable him to insist that any vagueness inherent in the PPA does not undermine its soundness (see Fine 2020; Keefe 2020). As an ethicist, though, I am inclined to think that the problem of specificity is a serious challenge to cultural relativism and therefore that he should not go this route.

Alternatively, he could attempt to show that the PPA does not involve a sorites-type paradox by devising a way to distinguish between trivial evils and significant evils. If this can be done, it would certainly be worth doing.

Finally, there are a number of strategies that philosophers have devised to defend their arguments against the accusation that they involve a problematic sorites-type paradox. Oms and Zardini's book referenced above discusses at least a dozen of these (Oms and Zardini 2019). Sterba might decide that one of these applies to and saves his argument.

**Funding:** This research received no external funding.

**Institutional Review Board Statement:** Not applicable.

**Informed Consent Statement:** Not applicable.

**Data Availability Statement:** Not applicable.

**Conflicts of Interest:** The author declares no conflict of interest.

### Notes

[1]    Several anthologies provide the key texts on the POE. See (Peterson 2016; Larrimore 2001). Meister and Moser (2017) is a nice collection of contemporary essays on the subject.

[2]    I was not able to find an actual definition of evil in Sterba's book, but the examples that he gives and the general discussion of evil in this book leads me to believe that he intends the word to refer to something like "unjustified suffering or loss."

[3]    Sterba is professor of ethics and political philosophy at the University of Notre Dame. His CV is provided on the University of Notre Dame website: https://philosophy.nd.edu/people/faculty/james-sterba/ (accessed on 1 November 2022). Of the 36 books that he has authored and edited, all but the last two are on ethics and/or political philosophy. The last two are on the POE.

[4]    Sterba was president of the Central Division of the APA in 2007–2008, president of the North American Society for Social Philosophy from 1990 to 1995, president of Concerned Philosophers for Peace in 1990 and 1991, and president of the International Association for Philosophy of Law and Social Philosophy, American Section from 1987 to 1989. He is a Fulbright Scholar and has been a visiting professor at the University of San Francisco, University of California at Irvine, University of Santa Clara,

5     Wuhan University (Wuhan, China), and Peking University (Beijing). His many grants, awards, and publications are listed here: https://philosophy.nd.edu/people/faculty/james-sterba/ (accessed on 1 November 2022).

5     According to Sterba, one of the first steps in this new activity was the procurement of a 2013–2014 grant from the Templeton Foundation funding research and two conferences on the Problem of Evil. He writes, "It was only in 2013 after receiving a grant from the John Templeton Foundation that I was able to fully bring my years of working in ethics and political philosophy to bear on the problem of evil" (Sterba 2019, p. 194).

6     Personal conversation with Sterba, University of Notre Dame, 6 November 2021, and (Sterba 2019, pp. v, 1).

7     A brief account of this is found in the preface to (Sterba 2019, p. v).

8     Sterba interprets Romans 3:8 as providing this principle (Sterba 2019, p. 48). I'm inclined to disagree with this interpretation, but Sterba's argument is not dependent on his interpretation of this passage and hence our disagreement on the interpretation of Rom. 3:8 is irrelevant to this article.

9     Sterba sometimes uses the term "horrendous evils" instead of "significant evils", and sometimes puts the two together. He uses Marilyn Adams' definition of horrendous evils as "[those evils] the participation in which (that is, the doing or suffering of which) constitutes prima facie reason to doubt whether the participant's life could (given their inclusion in it) be a great good to him/her on the whole," (Sterba 2019, p. 14).

10    One caveat is in order here: it seems to me that Sterba might be able to argue (or might be attracted by the argument) that the cumulative weight of the tremendous number of past and present non-significant evils is sufficient to serve as evidence for the non-existence of God.

11    Between chp. 3 and chp. 5 lies a chapter in which Sterba significantly elaborates the PPA.

12    In chp. 8 Sterba discusses natural evil, but this constitutes an application or intensification of his argument from evil rather than a response to his argument.

13    See the many articles on Sterba's POE in volumes 12 and 13 of *Religions* (published in 2021 and 2022), including (Sterba 2021). See also *International Journal for Philosophy of Religion* 87:3, June 2020. This entire issue of *IJPR* is devoted to debating Sterba's version of the POE.

14    On p. 125 of *Is a Good God* he states that there are only two such types of goods, but as he discusses them, he subdivides each into two, thus yielding a total of four. On pp. 185–88 of his conclusion he reiterates this fourfold categorization.

15    These examples are mine rather than Sterba's because he provides no examples.

16    Sterba mentions Mackie many times in his book and includes one of Mackie's books in his bibliography: (Mackie 1982).

17    God > ~evil; ~(~evil); therefore ~God.

18    This is my reconstruction (and condensation) of Sterba's argument, but something similar is found in (Sterba 2019, pp. 189–90). The term "omniscient" is ommitted because Sterba omits it. He does not explain why he has done so, but he is clearly talking about an Anselmian God and hence I believe we should take omniscience to be assumed. However, even if we do not assume actual omniscience, clearly Sterba wants us to assume that God is aware of the horrendous evils that motivate his argument.

19    Interestingly, here Sterba seems to reject the Pauline Principle.

20    On p. 23 he briefly explains the problem and implies that it undermines cultural relativism. On p. 25 he implies that the problem of specificity is one among a number of "difficulties" of relativism, thus indicating that he considers this to be a significant objection to relativism.

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
