# Peer review of "Sterba’s Problem of Evil vs. Sterba’s Problem of Specificity: Which Is the Real Problem?"

_religions, doi:10.3390/rel13111073_

Round 1
Reviewer 1 Report
In the "Conclusions," the Author writes: "Other philosophers have already championed this position, and it would enable him to insist that any vagueness inherent in the PPA does not undermine its soundness." I think it would be useful to indicate who these "other philosophers" are. As a reader of the text, I would love to find out.
The author of the text refers to a small number of sources. This is, I believe, due to the fact that the article is a polemic against a very specific argument proposed by Sterba.
The text is very interesting and excellently written.
Author Response
Dear Religions reviewer, 31 October, 2022
I thank you for the excellent advice that you have provided to me regarding improvements to the manuscript that I submitted to your journal. I have made the revisions that you recommended:
- Cite some authors who have embraced vagueness. This should be a footnote to my statement, “One option for saving Sterba’s argument would be for him to repudiate his ‘problem of specificity’ argument against moral cultural relativism and argue that vagueness is a ubiquitous feature of human discourse that does not present a serious problem to philosophical arguments and theories. Other philosophers have already championed this position, and it would enable him to insist that any vagueness inherent in the PPA does not undermine its soundness.”
- I have added the following footnote: 33 See Kit Fine, Vagueness: A Global Approach (New York: Oxford University Press, 2020); Rosanna Keefe, Theories of Vagueness (New York: Cambridge University Press, 2020).
- Add more sources to the text.
- I have added sources to several footnotes and have added these same sources to the reference list, which now looks like this:
- Asgeirsson, Hrafn. 2019. The Sorites Paradox in Practical Philosophy. In The Sorites Paradox. Edited by Sergi Oms and Elia Zardini. New York: Cambridge University Press, pp. 229-45.
- Fine, Kit. 2020. Vagueness: A Global Approach. New York: Oxford University Press.
- Keefe, Rosanna. 2020. Theories of Vagueness. New York: Cambridge University Press.
- Larrimore, Mark. 2001. The Problem of Evil: A Reader. Malden, MA: Blackwell Publishing.
- Mackie, J.L. 1982. The Miracle of Theism. Oxford: Oxford University Press.
- Meister, Chad, and Paul K. Moser, eds. 2017. The Cambridge Companion to the Problem of Evil. New York: Cambridge University Press.
- Oms, Sergi, and Elia Zardini, eds. 2019. The Sorites Paradox. New York: Cambridge University Press.
- Peterson, Michael L. 2016 The Problem of Evil: Selected Readings, 2nd ed. Notre Dame, IN: University of Notre Dame Press.
- Sterba, James P. 2021. Sixteen Contributors: A Response. Religions 12:7, 536.
- Sterba, James P. 2013. Introducing Ethics for Here and Now. Upper Saddle River, NJ: Pearson Education.
- Sterba, James P. 2019. Is a Good God Logically Possible? Cham, Switzerland: Palgrave Macmillan.
The resulting article is clearly better than my original submission. Thank you for sacrificing your time to help me improve my work.
Sincerely,
The Author
Reviewer 2 Report
Specific Comments
The following comments are listed in order of section and with reference to line numbers.
Abstract
· Why does the author bring up “Anselmian God” in the abstract but it never appears throughout the paper?
o The invocation of Anselm, while understandable, is odd if the grammar is not harnessed throughout the rest of the paper.
1. Introduction
· Lines 17-25
o This paragraph references those favoring the argument for the nonexistence of God from the problem of evil and those against such a position, but no footnote offers “who these people are.”
§ It would be helpful for a footnote referencing a “history of research” for this issue, even if only to point to a source that already does this well.
o In line 19, the attributes listed are “omniscient, omnipotent, and omnibenevolent,” but in lines 24-25, Sterba is noted to only deal with “omnipotent and omnibenevolent.”
§ Consistency in the attributes of God invoked in the conversation at hand would help the reader follow the argument throughout.
§ Otherwise, at the outset, one may query: would Sterba’s position be strengthened with the addition of “omniscient,” thereby nullifying this article’s critique? Who deals with “omniscient” in this regard (which the “history of research” footnote could suggest)?
§ Since omniscience doesn’t appear in the rest of the article, it could be safely omitted or addressed in a brief footnote.
· Lines 26-33
o In line 26, for clarity, add “In his prior work on ethics…”
§ Since the article’s methodology depends on utilizing Sterba’s prior work on ethics to dialog and to critique Sterba’s 2019 work, the insertion of “prior” signals to the reader, even if indirectly, the methodology of the article’s argument.
o Also, while line 26 mentions Sterba’s “work on ethics,” there is no footnote in the entire paragraph elucidating what “works” are being referenced here.
§ In fact, outside of a citation from a “personal conversation with Sterba” in footnote 3, there is not another source by Sterba until footnote 19 when Sterba’s “Introducing Ethics for Here and Now…2013” is cited (which is referenced again in footnotes 20 and 21).
2. Sterba’s Argument
· Line 34
o The opening line states: “I’ll begin by explaining Sterba’s argument.”
§ The problem is, in light of what precedes this line, there are two arguments that could be referenced: [1] The argument against moral cultural relativism or [2] the argument from evil.
§ Greater specificity is needed to determine “which of Sterba’s argument” is being referenced to.
· Lines 35-37
o The author offers a summary of Sterba’s expertise without any references for the reader in the footnotes:
§ [1] “main areas of work are moral and political philosophy” (lines 35-36)—what references support this?
§ [2] “His work in this area is well known and highly respected” (line 36)—what reference(s) support this? By whom?
§ [3] “In recent years he has also begun working on issues in philosophy of religion.”—what reference(s) support this?
o Summarizing a person’s work without any reference to their work even in footnotes is difficult for the reader to blindly trust and even seems like a bit of a distraction.
§ In fact, footnote 4 is a good example of what would be helpful here.
§ After referencing his work on the “problem of evil” with two conferences a Notre Dame, a published anthology, several articles, and conference presentations, you offer in footnote 4: “A brief account of this is found in the preface to Sterba, Is a Good God, v.”
· Line 42
o If the acronym POE is going to be used for the “Problem of Evil,” this should be introduced the first time this appears in the text.
o So, for example, the first time “problem of evil” is referenced is in line 18, which should be rendered as: “…argument from the “problem of evil” (POE).”
· Lines 49-52
o Two issues regarding clarity:
§ [1] The opening apposition is a bit wordy and redundant given the previous paragraph’s introduction of the theory (esp. lines 37-39).
§ [2] In lines 49-50, the author introduces the “Pauline Principle,” which in lines 50-52, the author explains both the definition and name of the “Pauline Principle.” However, in between the two sentences centered on “the Pauline Principle” the author introduces a related ethical principle of “the Principle of Double Effect” (line 50).
· There is no discernible reason to separate the sentences on the “Pauline Principle,” nor the necessity of connecting it with “the Principle of Double Effect”—especially when no definition of “the Principle of Double Effect” is offered.
· Footnote 8
o Awkward wording to introduce Marilyn Adam’s definition of horrendous evils:
§ “He uses Marilyn Adams’ definition of horrendous evils as evils “the participation in which…”
o It seems like what is intended is something like this:
§ “He uses Marilyn Adams’ definition of horrendous evils as “the participation [in evils] in which…”
· Lines 75-77
o Here, as earlier, the author mentions other pertinent works in this conversation without referencing them in any discernible manner in the body nor in a footnote.
§ “Other philosophers have analyzed the strengths and weaknesses of Sterba’s contributions to each of these and their analyses have been published in this and other periodicals.”
o Insert a footnote that offers, at least, a sampling of the other philosophers.
· Lines 140-142
o Here, as earlier, the author mentions other pertinent works in this conversation without referencing them in any discernible manner in the body nor in a footnote.
§ “Others have applied lessons from metaphysics and epistemology to issues in philosophy of religion with great success, so I’m inclined to think that this idea is a good one.”
o Insert a footnote that offers, at least, a sampling of the “others.”
3. My Argument
· Lines 146-151
o It is not entirely clear what the author is doing here:
§ Is the author introducing Sterba’s use of the Pauline Principle in his ethics book to demonstrate his inconsistency in applying it to utilitarian ethics and how he applies it to the problem of evil?
§ The author introduces in lines 149-150 a new principle (“the problem of specificity”) that, heretofore, has not appeared (save the title). Why? What is the connection to the previous discussion?
o Possible Solution: In light of line 239, the author could be more deductive in this opening paragraph stating: “Sterba’s reliance on the PPA to disprove the existence of God reduces his overall argument into incoherent ambiguity. This is illustrated through is earlier use of the principle of specificity.”
4. Conclusion
· This conclusion offers a couple of ways that Sterba could respond to this article’s critique. It is strange, in that, the argument is not so much summarized (as that is done in lines 256-270).
o Although this is a charitable way to end, it does seem to posture this more as a “book review” than an article.
o Indeed, the last sentence (and paragraph) ends the article rather abruptly, leaving the argument of the article a bit behind.
References
Never lists Sterba’s Is a Good God Logically Possible? Work (which is the center of this article).
Author Response
Dear Religions reviewer, 31 October, 2022
I thank you for the excellent advice that you have provided to me regarding improvements to the manuscript that I submitted to your journal. I have made almost all of the revisions that you recommended. I do have reservations about a small number of them, specifically 4, 12, and 15. Furthermore, you were not fully satisfied with the way the article concludes but you did not request any specific change to the conclusion. I am satisfied with the conclusion and wish to leave it unchanged.
The following is my summary of the changes that you requested together with my explanations of the changes that I have implemented. The resulting article is clearly better than my original submission and I thank you for sacrificing your time to help me improve my work.
Sincerely,
The Author
Requested Changes:
- Why is the Anselmian God mentioned in the preface but not in the paper?
- My reply: This term is a commonly-used synonym for “the God of classical theism.” The term Anselmian is used on line 6 and also on line 62, and the phrase “the God of classical theism” is occurs on lines 216, 241, and 257. Since Sterba occasionally uses the former term but never uses the latter, I’ve replaced the latter with the former throughout. I’ve also specified on line 62 that “Anselmian” means omniscient, omnipotent, and omnibenevolent.
- The first paragraph mentions past work on the POE in a very general way. Cite some specifics.
- My reply: A footnote has been added citing several sources.
- Line 19 lists omniscience, omnipotence, and omnibenevolence, but the rest of the paper omits omniscience.
- My reply: Omniscience has been added to lines 24 and 62-3 and an explanation for its omission has been added to footnote 19.
- Line 26 says “In his work on ethics…” Make this “In his prior work on ethics…” to signify that I’m dealing with work published before the book to which I’m responding.
- My reply: Adding “prior” makes it sound like I’m contrasting Sterba’s work on ethics with the book currently under discussion, thus implying that the book currently under discussion is an ethics book. The book currently under discussion is not about ethics, so adding “prior” makes the sentence confusing. There’s no need to add the word “prior” since I’ve already explained that Sterba has a long history of work in the field of ethics and only recently turned to working on the POE. Hence after considering this change, I have decided to leave things the way they are.
- Also line 26: footnote needed for Sterba’s work on ethics.
- My reply: A footnote has been added for the specific book to which I am alluding.
- Line 34: Clarify that “I’ll begin by explaining Sterba’s argument” refers to his POE argument.
- My reply: Done.
- 35-6: Substantiate the claim that Sterba’s main areas of work are moral and political philosophy.
- My reply: I’ve added a footnote of corroboration.
- 36: Substantiate the claim that Sterba’s work is highly respected.
- My reply: I’ve added a footnote of corroboration.
- 36: Substantiate the claim that he has in recent years begun working on philosophy of religion.
- My reply: I have added corroboration via a footnote.
- 42: POE as an acronym for the Problem of Evil needs to be introduced on line 18.
- My reply: Done.
- 49-52 (and surrounding): The opening is wordy and redundant given the previous paragraph (esp. 37-39).
- My reply: This has been revised.
- 49-52: The reference to the principle of double effect is unneeded and should be deleted.
- My reply: The Principle of Double Effect is very well known in philosophical ethics. This succinct mention of it will help the reader to locate the less-known Pauline Principle in the correct philosophical tradition. As such, it is useful, it is not a distraction, and it should be retained.
- Footnote 8: Awkward wording (“horrendous evils as evils”)(better: “horrendous evils as “the participation [in evils] in which…”)
- My reply: This has been reworded.
- 75-77: Footnote some of the other analyses of Sterba’s work.
- My reply: Done.
- 140-2: Footnote some of the lessons from metaphysics and epistemology that have been successfully applied to PoR.
- My reply: such documentation is not needed for the argument advanced in this text and does not strengthen the argument of this text and therefore would be a distraction. Furthermore, such documentation is not needed because the target audience of this text is other scholars working in the areas of philosophy and religious studies and such professionals will already be familiar with this material.
- 146-151: The reviewer does not seem to follow the argument here, though s/he doesn’t explain why. The reviewer offers a possible rewording that changes the argument.
- My reply: The paragraph comprised of lines 148 through 154 is a bridge from the preceding section of the article to the following paragraphs, which develop my objection to Sterba. It begins by succinctly showing that Sterba’s use of the Pauline Principle in his POE is rooted in his use of the Pauline Principle in ethics, consonant with Sterba’s theory that strategies used successfully in ethics can be applied fruitfully to issues in philosophy of religion. Then the paragraphs introduces the concept from ethics that I plan to apply to the POE, which is also found in Sterba’s work on ethics – indeed in the very same book. This reflects my agreement with Sterba that strategies used successfully in ethics can be applied fruitfully to issues in philosophy of religion. I have modified lines 153 to 155 in order to make this transition clearer.
- Conclusion: The reviewer seems dissatisfied with the conclusion….
- My reply: My article ends the way it does because I have written it in the spirit of constructive criticism. I do not wish to present my argument as if it is the final word on this issue.
- References: The reviewer was not able to find Is a Good God Logically Possible? in the reference list.
- My reply: This book is in fact listed, but unfortunately MBTI has conflated it with the reference that precedes it. Since I have added other texts to the reference list, I have deleted the original reference list in its entirety and have replaced it with an updated one in which three of Sterba’s books are listed, including Is a Good God Logically Possible?

Round 2
Reviewer 2 Report
I have reviewed the author's responses and revised paper. I am content with the modifications. Well done.